# A comparative study on the use of microscopy in pharmacology and cell biology research

**Agatha M. Reigoto**⊛, **Sarah A. Andrade**⊛, **Marianna C. R. R. Seixas**⊛, **Manoel L. Costa**, **Claudia Mermelstein** (ID) *

Instituto de Ciências Biomédicas, Universidade Federal do Rio de Janeiro, Rio de Janeiro, Brazil

⊛ These authors contributed equally to this work.
* mermelstein@ufrj.br

**Data Availability Statement:** All relevant data are within the manuscript and its Supporting information files.

**Funding:** This work was supported by Conselho Nacional de Desenvolvimento Científico e

## Abstract

Microscopy is the main technique to visualize and study the structure and function of cells. The impact of optical and electron microscopy techniques is enormous in all fields of biomedical research. It is possible that different research areas rely on microscopy in diverse ways. Here, we analyzed comparatively the use of microscopy in pharmacology and cell biology, among other biomedical sciences fields. We collected data from articles published in several major journals in these fields. We analyzed the frequency of use of different optical and electron microscopy techniques: bright field, phase contrast, differential interference contrast, polarization, conventional fluorescence, confocal, live cell imaging, super resolution, transmission and scanning electron microscopy, and cryoelectron microscopy. Our analysis showed that the use of microscopy has a distinctive pattern in each research area, and that nearly half of the articles from pharmacology journals did not use any microscopy method, compared to the use of microscopy in almost all the articles from cell biology journals. The most frequent microscopy methods in all the journals in all areas were bright field and fluorescence (conventional and confocal). Again, the pattern of use was different: while the most used microscopy methods in pharmacology were bright field and conventional fluorescence, in cell biology the most used methods were conventional and confocal fluorescence, and live cell imaging. We observed that the combination of different microscopy techniques was more frequent in cell biology, with up to 6 methods in the same article. To correlate the use of microscopy with the research theme of each article, we analyzed the proportion of microscopy figures with the use of cell culture. We analyzed comparatively the vocabulary of each biomedical sciences field, by the identification of the most frequent words in the articles. The collection of data described here shows a vast difference in the use of microscopy among different fields of biomedical sciences. The data presented here could be valuable in other scientific and educational contexts.

## Introduction

How important is it to look and study cells and tissues at the microscope in biomedical research? Which microscopy techniques are used in different fields of biomedical research? Since cells are within the micrometer scale, understanding the cellular basis of human health

Tecnológico (CNPq/Brazil) number 407331/2018-2 to C.M., and Fundação Carlos Chagas Filho de Apoio à Pesquisa do Estado do Rio de Janeiro (FAPERJ) number E-26/202.920/2019 to C.M. The funders had no role in study design, data collection and analysis, decision to publish, or preparation of the manuscript.

**Competing interests:** The authors have declared that no competing interests exist.

**Abbreviations:** BJP, British Journal of Pharmacology; BrF, bright field microscopy; CEL, Cells; Conf, confocal microscopy; CrEM, cryogenic-electron microscopy; DIC, differential interference contrast microscopy; DOI, digital object identifier; EM, electron microscopy; Flu, fluorescence microscopy; FP, Frontiers in Pharmacology; JBC, Journal of Biological Chemistry; JCB, Journal of Cell Biology; JCS, Journal of Cell Science; JPP, Journal of Pharmacy and Pharmacology; Live, live microscopy; NF-KB, nuclear factor kappa-light-chain-enhancer; OM, optical microscopy; Pha, phase contrast microscopy; PNAS, Proceedings of the National Academy of Sciences; SEM, scanning electron microscopy; SRes, super-resolution microscopy; TEM, transmission electron microscopy.

and disease requires the spatial resolution of microscopy [1]. Light and electron microscopy are among the major techniques used to study cellular structure and function [2]. The first microscopes were invented in 1600, and they led to the first observation of cells by Robert Hooke in 1665 [3] which in turn led to the elaboration of the Cell Theory. In the last forty years microscopy has undergone a revolution from largely qualitative observations in fixed cells to high-throughput quantitative data in live cells [4]. Today there are several different microscopy techniques to improve the visualization of fixed or live cells. Different techniques, including the light-based bright field, phase contrast, differential interference contrast (DIC), polarization, fluorescence, and confocal microscopy, and the electron-based scanning and transmission microscopy have different advantages. Both phase contrast and DIC are optical microscopy techniques used to enhance the contrast of unstained and transparent samples, including live specimens. The easiness of the phase contrast makes it a perfect match to cell cultures, while DIC achieves higher resolutions but is more labor-intensive. Polarization microscopy also enhances contrast without stain, but depends on birefringent materials, such as collagen, cellulose, myofibrils, and microtubules, and is not applicable to any molecule [5]. Fluorescence achieves very high signal to noise ratio [6], but is usually dependent on labelled antibodies or probes, or in the expression of proteins tagged with fluorescence molecules, such as green fluorescent protein (GFP) and its derivatives [7]. Confocal microscopy [8], usually based on the use of a pinhole to reject out-of-focus fluorescence, is necessary for the observation of thick fluorescent specimens, up to the size of a whole zebrafish larvae. Super-resolution microscopy methods [9] bypass the resolution limit of 0.2 um established by Ernst Abbe in the 19th century either using an interfering pattern (SR-SIM), or a de-excitation laser (STED), or can be based on the localization of fluorochromes (STORM). Emerging developments in live-cell microscopy and fluorescent labeling have begun to open unique opportunities to reveal the dynamics of biological systems with high spatio-temporal details [10, 11]. Electron microscopy achieve higher resolution than optical microscopy because the wavelength of electron is dependent on the voltage applied to the beam, and can be much smaller than the wavelength of light. The transmission electron microscopy (TEM) depends on very thin sections and contrast based on molecules, such as osmium and lead [12]. TEM uses magnetic lenses in a way similar to the transillumination of optical microscopy, while scanning electron microscopy (SEM) uses an electron beam to scan the metal-coated surface of a non-sectioned specimen. In general electron microscopy is capable of much more detail than optical microscopy, but with much more work and equipment cost. Cryogenic-electron microscopy is a powerful technique that recently emerged in structural biology, capable of delivering high-resolution density maps of macromolecular structures [13].

One of the areas in which microscopy could have made an impact is pharmacology. Imaging methods allow pharmacology researchers to address a vast number of biological questions, such as, *in vivo* analysis of the effects of specific drugs or molecules in a cell and/or a tissue's morphology and physiology, nanoparticle-cell interactions, intracellular redox chemistry, mitochondrial physiology, structural determination of new drugs, among others [14–18]. Although some authors have pointed out the importance of microscopy in drug discovery [19–21], the question remains if microscopy techniques are used in pharmacology research.

To test how important microscopy is to specific biomedical research fields, we decided to gather data on the use of microscopy in published articles in pharmacology, cell biology and other fields of biomedical sciences. Analysis of data obtained from published articles can be a useful tool to obtain a comprehensive view of specific research fields [22]. Our approach was to quantitatively analyze (i) the overall use of microscopy in recently published articles in pharmacology and, comparatively, in cell biology journals and other related fields, (ii) the use of different techniques of optical and electron microscopy in these articles, (iii) the correlation

between the use of microscopy and cell cultures, and (iv) the differences in the vocabulary of the articles based on the relative frequency of words in their titles.

## Materials and methods

### Characteristics of material and description of the process

In this study we analyzed the use of microscopy in articles published in eight leading scientific journals from the pharmacology, cell biology, biochemistry, and general biomedical sciences fields (Table 1). The selected journals were British Journal of Pharmacology (BJP), Journal of Pharmacy and Pharmacology (JPP), Frontiers in Pharmacology (FP), Journal of Cell Biology (JCB), Journal of Cell Science (JCS), Cells (CEL), Journal of Biological Chemistry (JBC) and Proceedings of the National Academy of Sciences (PNAS). Since microscopy is a very dynamic field, we selected the year of 2019 to provide data related to recent articles in which newly established microscopy methods could have been used. BJP (impact factor of 7.7) is published by the British Pharmacological Society and was established in 1946. JPP (impact factor of 2.4) is the official journal of the Pharmaceutical Society of Great Britain, was established in 1870 and obtained its current title in 1949. Compared to BJP and JPP, FP (impact factor of 4.2) is a relatively new pharmacology journal (established in 2011) and is the first most cited open-access journal in the pharmacology field. JCB (impact factor of 8.8) is published by Rockefeller University Press and was established in 1962. JCS (impact factor of 4.6) was established in 1853 and it is published by The Company of Biologists. Compared to JCB and JCS, Cells (impact factor of 4.4) is a recently new journal (the first articles were published in 2012) and it is an open-access journal in the cell biology field. JBC (impact factor of 4.2) was established in 1905 and since 1925 it is published by the American Society for Biochemistry and Molecular Biology. JBC covers research in areas of biochemistry and molecular biology. PNAS (impact factor of 9.4) is the official journal of the National Academy of Sciences (USA) and one of the world's most cited and comprehensive multidisciplinary scientific journals. We analyzed the same number of articles (200) from all the eight journals and to obtain this equal number we included all the articles published by them in 2019 and when necessary with articles from the beginning of 2020 (Table 1). We excluded from our analysis all the Reviews and Editorial Comments from these journals, and therefore, only original articles were included.

For the analysis of the use of microscopy, we examined the following data from each article: title, abstract, methodology section, figures, figure legends and results section. We analyzed in each article the use of 8 types of optical microscopy techniques: bright field (BrF), phase contrast (Pha), differential interference contrast (DIC), polarization (Pol), conventional fluorescence (Flu), confocal fluorescence (Conf), super resolution (SRes), and live cell imaging (Live); and 3 types of electron microscopy techniques: transmission (TEM), scanning (SEM) and

**Table 1. Information on the biomedical sciences journals used in this study.**

| Journal's name | Journal's abbreviation | Number of articles analyzed |
|---|---|---|
| British Journal of Pharmacology | BJP | 200 |
| Journal of Pharmacy and Pharmacology | JPP | 200 |
| Frontiers in Pharmacology | FP | 200 |
| Journal of Cell Biology | JCB | 200 |
| Journal of Cell Science | JCS | 200 |
| Cells | CEL | 200 |
| Journal of Biological Chemistry | JBC | 200 |
| Proceedings of the National Academy of Sciences | PNAS | 200 |

cryo-EM (CrEM). All the data (article's title, DOI, use of microscopy, use of cell culture, number of figures with microscopy) was plotted in a spreadsheet and graphs were generated using Microsoft Excel™ software. We did not include some microscopy methods, such as Light Sheet, Raman, Atomic Force, Intravital, Dark Field, and High-Content Screening, because they were used by a very small number of articles.

### Data mining and textual analysis

We also analyzed the frequency of words that appear in the title of the articles from the eight selected journals. Wordle™ software (freely available at http://www.wordle.net and created by Jonathan Feinberg) [23] was used to generate a list of words with their relative frequencies, and to generate "word clouds". The clouds give greater prominence to words that appear more frequently in the source text, i.e., more frequent words appear with larger letters and in a colored gradient. We used the following parameters to generate word clouds: remove common English words and numbers (e.g., "and", "all", "to", "at"), make all words lower-case, Telephoto font type, rounded edges, kindled color, horizontal layout. We only used words that appeared at least three times in all the titles of the 200 articles from each journal. The list of words was manually edited to remove plural words (e.g., "cells" changed to "cell"), different spellings for some words (e.g., "signaling" changed to "signalling"), and symbols (e.g., "NF-kB" changed to "NF-KB").

## Results and discussion

### Comparative analysis of the use of microscopy in biomedical journals

To evaluate the importance of morphological studies in different fields in biomedical sciences, we analyzed the relative use of microscopy in articles published in eight leading scientific journals from pharmacology, cell biology and other biomedical sciences fields. Our data shows (Fig 1) that microscopy was used by almost all articles from two cell biology journals (97% in JCB and JCS) and highly frequent in a new cell biology journal (75% in CEL). In a different way, pharmacology journals used microscopy in approximately half of the articles (49% in BJP and FP, 51% in JPP). We found that approximately half of the articles in a biochemistry journal used microscopy (55% in JBC), while only 36% of the articles in a multidisciplinary journal (PNAS) used microscopy. Optical microscopy (OM) was much more used than electron microscopy (EM). There are many possible reasons for the broader use of OM rather than EM: (i) because it is faster, (ii) possible to label multiple probes simultaneously, (iii) works on live cells, (iv) the biological structure is not damaged during the preparation, and (v) it can be fully quantitative. On the other side, EM is usually used where very high resolution is needed.

On pharmacology journals, OM was used in 45–59% articles, whereas cell biology journals used 75–97% of OM. Curiously, use of OM in articles from JBC (53%) was similar to pharmacology journals. Since PNAS is a general journal that publish articles in many subjects that are outside the scope of biological sciences, we analyzed both the percentage of OM use in all articles (32%) of the journal, as well as the percentage of OM use only in articles from the Biological Sciences section of PNAS (47%), which was within the range of OM use in the pharmacology field. It is important to point out that the Biological Sciences section of PNAS included a broad range of fields, such as Agricultural Sciences, Anthropology, Ecology, Evolution, Population Biology, Psychological and Cognitive Sciences, Sustainability Science, and Systems Biology, which only in rare occasions analyze cells and tissues and therefore do not demand the use of microscopy.

EM alone was rarely used in all journals (0–6%). EM was more used in two cell biology journals (20 and 23% in JCB and JCS), less frequent in CEL and PNAS (10%), and in

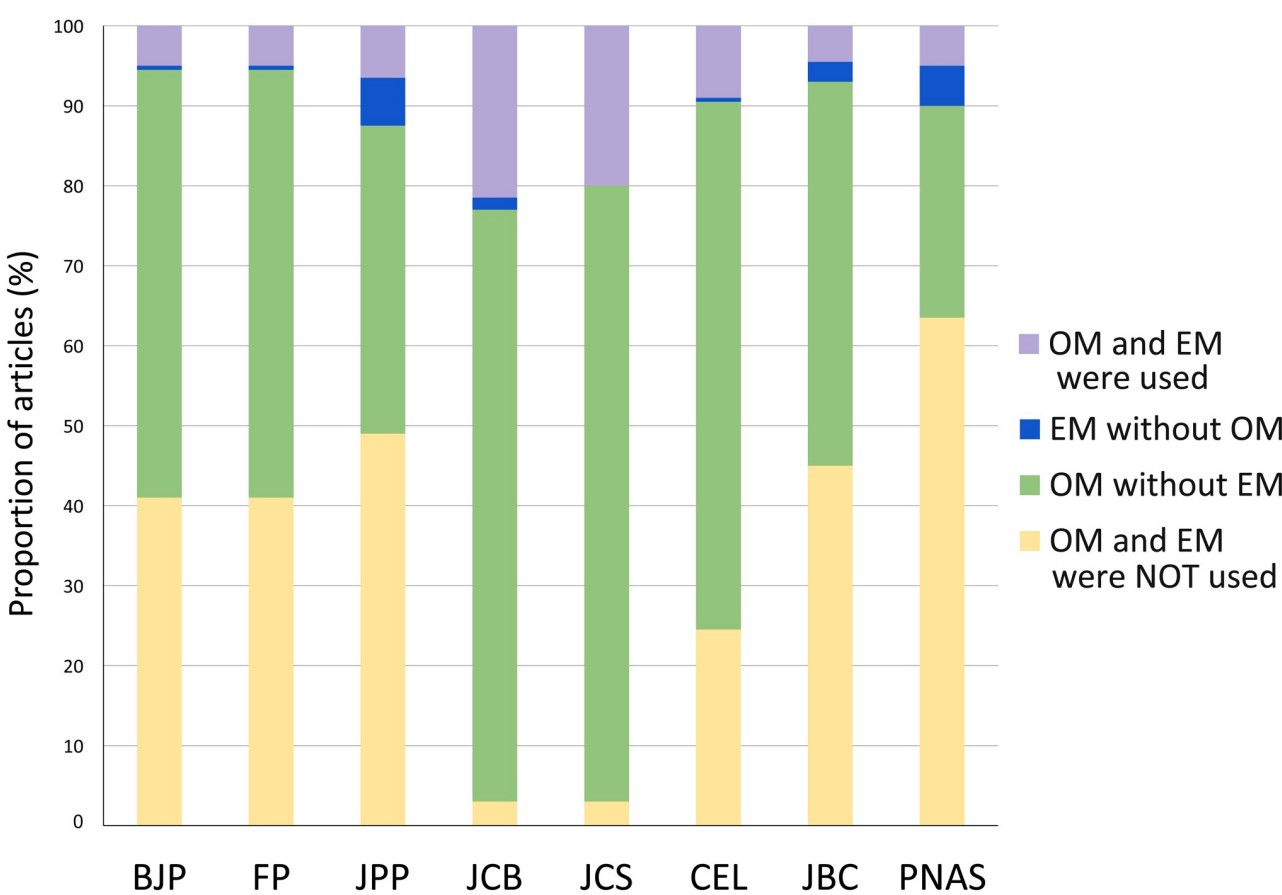

**Fig 1. Comparative analysis of the proportion of articles that uses microscopy techniques in biomedical sciences journals.** Analysis of the presence of optical (OM) and electron (EM) microscopy methodologies in all articles published in 2019 in the journals: British Journal of Pharmacology (BJP), Journal of Pharmacy and Pharmacology (JPP), Frontiers in Pharmacology (FP), Journal of Cell Biology (JCB), Journal of Cell Science (JCS), Cells (CEL), Journal of Biological Chemistry (JBC) and Proceedings of the National Academy of Sciences (PNAS). N = 200 articles analyzed in each scientific journal.

pharmacology and biochemistry journals (6–7%). Although some authors have pointed out the importance of microscopy in drug discovery and in other fields of pharmacology [19, 21], our data showed that microscopy techniques are not widely used in pharmacology research.

## Comparative analysis of the use of different optical and electron microscopy techniques

To further detail any difference in microscopy usage from different biomedical sciences fields, we analyzed in each article which OM and/or EM techniques were used (Fig 2). Overall, we found that the most frequent microscopy methods were bright field, conventional, and confocal fluorescence microscopy. In contrast, polarization microscopy was almost not used by articles in any journals, and DIC and SRes were almost only used in cell biology. Indeed, the use of polarization requires a birefringent material, while DIC and specially SRes are more labor-intensive and SRes is still a quite new and expensive microscopy technique.

Bright field was more frequently used by articles from pharmacology journals, while fluorescence was more frequently used by articles from cell biology journals, with the exception of the journal CEL, which used more bright field and less fluorescence. It is reasonable to assume

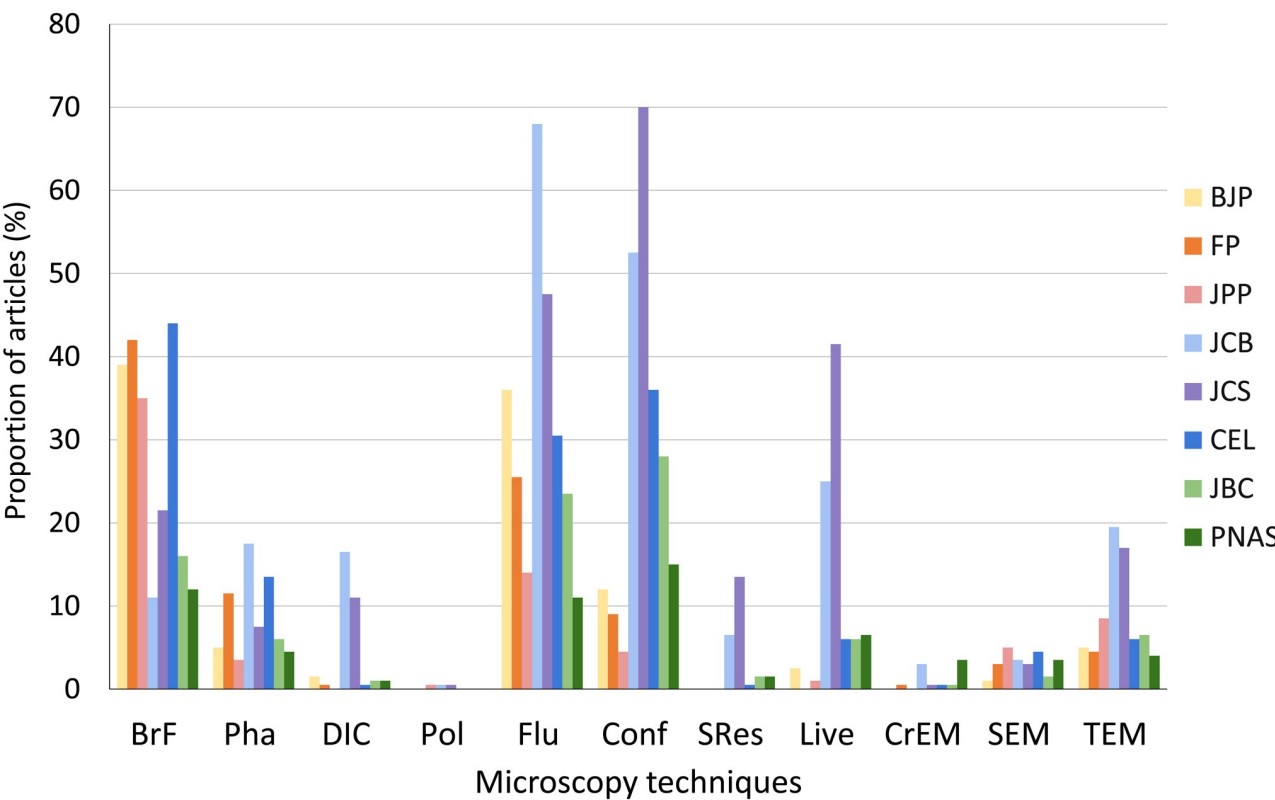

**Fig 2. Comparative analysis of the proportion of use of different optical and electron microscopy techniques in biomedical sciences journals.** We analyzed in each article the use of 8 types of optical microscopy techniques: bright field (BrF), phase contrast (Pha), differential interference contrast (DIC), polarization (Pol), conventional fluorescence (Flu), confocal fluorescence (Conf), super resolution (SRes), and live cell imaging (Live); and 3 types of electron microscopy techniques: transmission (TEM), scanning (SEM) and cryo-EM (CrEM). Data were collected from articles published in 2019 in the biomedical sciences journals: British Journal of Pharmacology (BJP), Journal of Pharmacy and Pharmacology (JPP), Frontiers in Pharmacology (FP), Journal of Cell Biology (JCB), Journal of Cell Science (JCS), Cells (CEL), Journal of Biological Chemistry (JBC) and Proceedings of the National Academy of Sciences (PNAS). N = 200 articles analyzed in each scientific journal.

that the use of bright field is mostly related to histological observations. We could speculate that pharmacology articles are more concerned with alterations in tissues caused by pharmacological treatments, while cell biology articles use fluorescence to focus on the distribution of specific molecules or organelles within cells. We cannot rule out the influence of traditional and historical methodological reasons for the different use of microscopy techniques in each field. Phase contrast microscopy was not frequently used by all the journals. This is remarkable, since several biomedical sciences papers use cell cultures, and phase contrast microscopy is a valuable technique to show cell morphology, cell culture density and many other characteristics of unlabeled live cells.

Fluorescence and confocal microscopy were frequently used in articles from two cell biology journals (53–70% in JCB and JCS), and less used in the journal CEL (31–36%). Conventional fluorescence (14–36%), and particularly confocal microscopy (5–12%), were infrequently used in pharmacology articles. Confocal optical microscopy is based on the suppression of out-of-focus fluorescence of three-dimensional (3D) structures. The complexity and the cost of confocal equipment varies, but they are usually quite high, and therefore this can be an important limitation on its use. Some differences in the use of confocal microscopy between cell biology and pharmacology journals could be attributed to the use of bright field in histological sections, as opposed to the optical sections obtained with confocal microscopy.

Nevertheless, one must assume that pharmacological studies do not deal with 3D-structures as much as cell biology.

Interestingly, live cell microscopy was mostly used by the cell biology journals JCS (42%) and JCB (25%), while the other cell biology journal CEL and all other journals used very rarely (0–6%). A lot of work and some specialized equipment are necessary to use live cell microscopy, but it seems that the results that have being obtained are valued for some cell biology journals.

TEM was not frequently used in publications (less than 20% of articles from all journals) but was the most used EM technique in all journals. We observed that the frequency of use of SEM is similar in all journals (1–5%), while TEM was much more frequently used by the cell biology journals JCB (20%) and JCS (17%). TEM and SEM are both electron microscopy methods that traditionally have been used in the biomedical sciences for detailed structural subcellular analysis of cells and tissues (TEM) and for the characterization of cell and tissue surface topography (SEM). One can argue that TEM and SEM are not widely used because they are labor and cost-intensive techniques that usually depend on specific institutional facilities. It is worth mentioning that TEM is an essential tool for the detection and analysis of the localization and ultrastructure of molecules/structures at the nanoscale level: the most up-to-date electron cryomicroscopy methods are now close to atomic resolution [24]. For example, TEM is widely used to visualize viruses in cells and tissues; and therefore, could be expected to be found more frequently in studies on the effects of specific drugs targeted to different viruses.

We also analyzed the use of Cryogenic-electron microscopy (CryEM). Even though CryEM is considered as a revolutionary technique for determining the 3D shape of macromolecules, and therefore has been described as an important technique for drug discovery (Renaud et al. 2018), our analysis revealed that CryEM was only found in articles published in JCB (3%) and PNAS (3.5%), and almost not detected in pharmacology articles. Cryo-EM is most likely being used for basic-science studies, which could explain the low-percentage use.

## Comparative analysis of the combined use of different microscopy techniques

Next, we asked if the different microscopy techniques were used in combination or isolated. Therefore, we plotted the number of microscopy techniques used concomitantly in each article. As described before, nearly 50% of the articles published in all pharmacology journals (BJP, FP and JPP), and in the biochemistry journal JBC employed no microscopy methods (Fig 3). Remarkably, 64% of the articles from PNAS did not use any microscopy, which could be explained by the variety of research areas it covers since it is a multidisciplinary journal. JCB and JCS articles more frequently combine 2 microscopy techniques, while CEL more frequently uses only a single microscopy technique. The cell biology journals used combinations of 2 to even 6 microscopy methods in the same article. No article from any journal used more than 6 microscopy methods from the total of 11 methods that we analyzed. It is worth mentioning that these microscopy techniques have different advantages and may complement each other, and therefore their combined use could improve the article and should be stimulated. S1 File shows the frequency of each of the combinations of microscopy techniques that were used simultaneously in the articles. The combination use of 6 techniques were used in only 2 articles (from JCB) and these were: BrF + DIC + Flu + Conf + Live + SEM, and BrF + Flu + Conf + SRes + Live + TEM. The combined use of 5 techniques were found in 15 articles and the most frequent one was BrF + Pha + Flu + Conf + Live. The concomitant use of 4 techniques were found in 37 articles and the most frequent one was DIC + Flu + Conf + Live. The use of 3

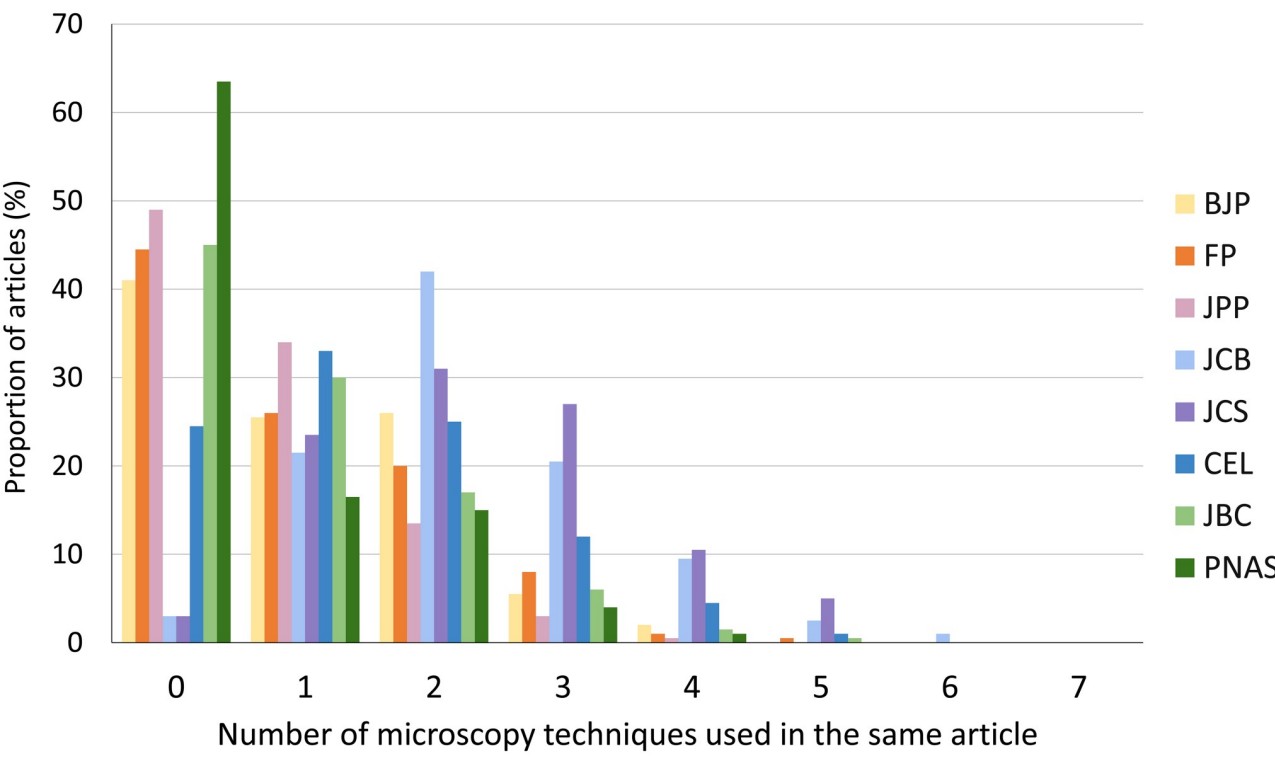

**Fig 3. Comparative analysis of the number of microscopy techniques used in articles from biomedical sciences journals.** Data were collected from articles published in 2019 in the biomedical sciences journals: British Journal of Pharmacology (BJP), Journal of Pharmacy and Pharmacology (JPP), Frontiers in Pharmacology (FP), Journal of Cell Biology (JCB), Journal of Cell Science (JCS), Cells (CEL), Journal of Biological Chemistry (JBC) and Proceedings of the National Academy of Sciences (PNAS). N = 200 articles analyzed in each scientific journal.

techniques was present in 41 articles and the most frequent ones were BrF + Flu + Conf, BrF + Pha + Flu, and Conf + SRes + Live. Finally, the combination of two techniques were detected in 28 articles and the most common ones were BrF + Flu, BrF + Conf, Flu + Conf, Conf + Live, Pha + Flu, Conf + TEM, and Flu + Live. Interestingly, although DIC and SRes were not frequently used techniques they were always used in combination with other techniques.

### Evaluation of the relative importance of microscopy to the article and correlation between the use of microscopy with the article's research theme

To further evaluate the impact of microscopy to each article, we analyzed the proportion of figures containing microscopy images in each article. These results showed that microscopy is usually one of the main techniques in cell biology but not in pharmacology: articles in pharmacology journals uses 16–25% of images with microscopy (Fig 4), while the cell biology journals JCB uses 76%, JCS 65% and CEL 35%. The biochemistry journal JBC uses on average 21% of microscopy in the figures of each article, while the multidisciplinary journal PNAS uses 18% of microscopy in each article. These data suggest that morphological information is more significant to research results in cell biology articles. Even though our analysis showed that nearly half of the articles from the pharmacology journals did not use any microscopy method (Fig 1), it is important to point out that a fraction of pharmacology research articles, on subjects such as biochemistry pathways and behavior, would not require microscopy. Therefore, we attempted to correlate the use of microscopy with the research theme of each article. Since the themes can be diverse, we focus on analyzing the use of cell cultures, since it is reasonable to

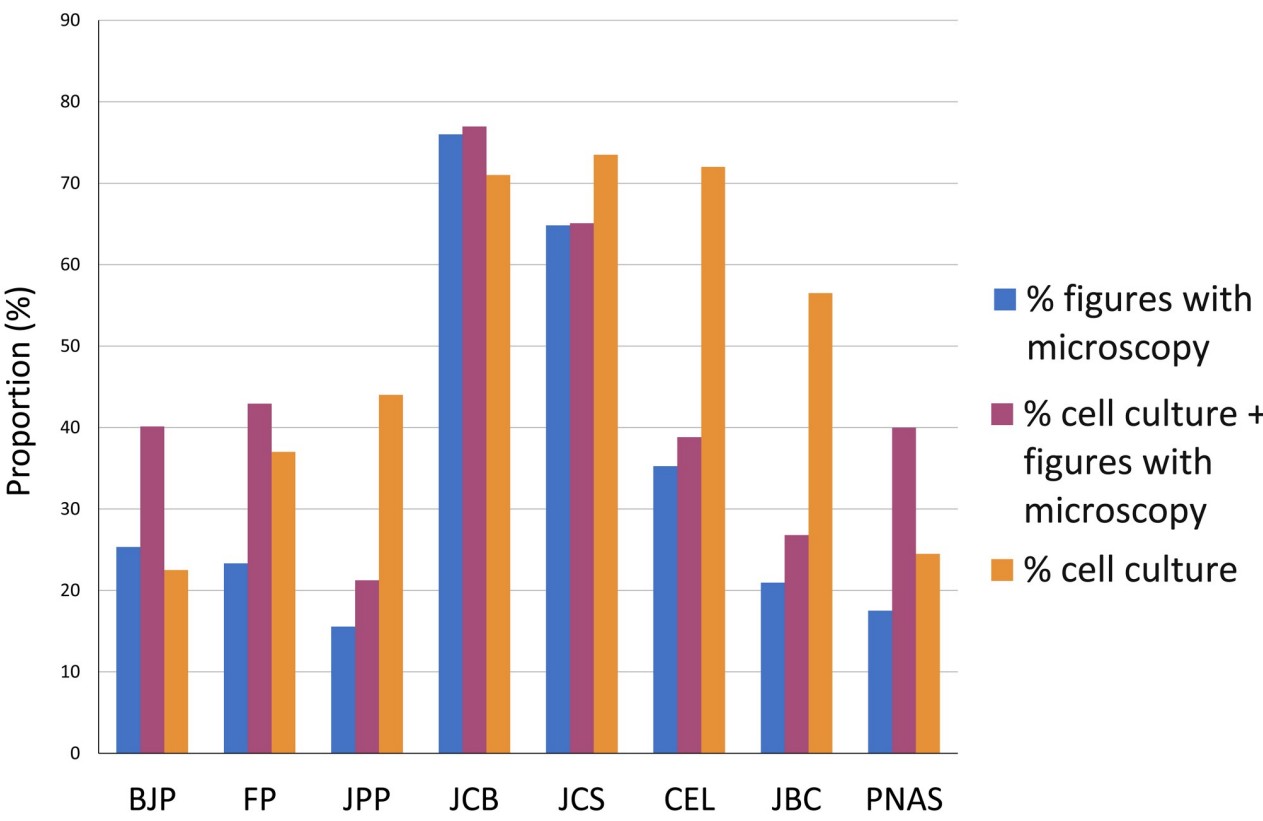

**Fig 4. Comparative analysis of the percentage of articles figures with microscopy and cell culture in biomedical sciences journals.** The percentage of figures containing microscopy images is shown in blue bars, the percentage of articles containing cell culture is shown in orange bars and the percentage of articles containing cell culture and microscopy is shown in magenta. Data were collected from articles published in 2019 in the biomedical sciences journals: British Journal of Pharmacology (BJP), Journal of Pharmacy and Pharmacology (JPP), Frontiers in Pharmacology (FP), Journal of Cell Biology (JCB), Journal of Cell Science (JCS), Cells (CEL), Journal of Biological Chemistry (JBC) and Proceedings of the National Academy of Sciences (PNAS). N = 200 articles analyzed in each scientific journal.

expect that microscopes are needed for the proper visualization of cells because of their dimensions. In articles that used cell cultures, we analyzed the percentage of microscopy figures in relation to the total number of figures of the articles. The results ranged, in the pharmacology journals, from 21% in JPP to 40–43% in BJP and FP; in cell biology articles 39% in CEL to 65–77% in JCS and JCB. In JBC articles that used cell cultures, 27% of the figures used microscopy, while in PNAS articles that used cell cultures 40% of the figures used microscopy. We conclude that on average, the number of figures using microscopy in articles that use cell culture is generally higher in the cell biology field that in pharmacology. Remarkably, we obtained similar percentage of use of microscopy within total figures combined with cell culture for JPP and JBC (21 and 27%, respectively), which is indicative of a frequent use of cell culture without a corresponding emphasis in cell imaging. Interestingly, although the total use of cell culture in PNAS was low (25%), the frequency of microscopy together with cell culture (40%) was much higher than in articles that do not use cell culture (18%).

### Analysis of vocabulary similarities in biomedical sciences

Finally, we hypothesized that the different usage of microscopy that we observed in the journals from different fields could be related to differences in the concepts used by each field. To

address this hypothesis, we would need a way of comparing the contents of each article. We therefore decide to analyze the relative use of words in the titles of the articles (S2 File). Since textual information is difficult to visualize quantitatively [25], we used the Wordle™ software to draw "word clouds" [23]. The "word clouds" give greater prominence to words that appear more frequently in the source text. Interestingly, "cell" was one of the three most frequent words found in the vocabularies from the titles of the articles from all journals (Fig 5). These results suggest that most of the articles published in these journals included analysis at the cellular level and therefore microscopy techniques would be a valuable tool for their studies. We observed that JCB and JCS showed a remarkably similar list of most frequent words: "cell", "regulation", "protein" and "signalling". This word usage suggest that these articles are concerned with cellular processes at the molecular level. We also observed a somewhat similar list of most frequent words between the three pharmacology journals: "cell", "receptor", "inhibitor" and "cancer", as well as similar model organisms ("mouse", "rat" and "human"). This pattern may result from the pharmacology vocabulary itself and the focus on diseases. The remaining journals (CEL, JBC and PNAS) showed a mixed list of most frequent words.

## General comments

During our analysis we noticed that many articles (i) did not describe at all some of the microscopy methodologies that were used, or (ii) described the microscopy methods with major errors, such as describing the use of one methodology when they used another one, and/or (iii) described the use of microscopy with important missing information on the use of microscopy methods. Remarkably, bright field and phase-contrast microscopy were the most under described microscopy techniques; most of the articles that have figures with these techniques did not mention or describe them in the methodology section. The collection of problems above described can lead to serious mistakes in the interpretation of the results presented in such papers, in the chance of an accurate repetition of these experiments and in how microscopy is perceived by researchers and their students.

## Conclusions

Here we established a procedure for the analysis of the content of articles that can be used for further similar studies in other areas and/or focusing on other aspects of research. In general, we could see several similar parameters between BJP and FP and between JBC and JCS. Interestingly, several parameters that we analyzed in JPP and CEL do not behave in a similar way to pharmacology and cell biology journals, respectively. The level of similarity between different journals of the same field should be further investigated.

Our analysis showed that nearly half of the articles from the pharmacology journals did not use any microscopy method, compared to the use of microscopy in almost all (97%) of the articles from the cell biology journals. We hope that our study will provide support for a critical evaluation of the impact of the use of microscopy in the pharmacology field and in biomedical sciences in general. Giving the advancement in the recent years in the microscopy field, allowing the high-resolution analysis of real-time dynamic processes in cells and tissues, we can conclude that the pharmacology research field could gain novel horizons by including new microscopy techniques in their studies.

## Supporting information

**S1 File. Frequency of the combinations of microscopy techniques used simultaneously in the articles from eight biomedical sciences journals.** The techniques analyzed were bright field (BrF), phase contrast (Pha), differential interference contrast (DIC), polarization (Pol),

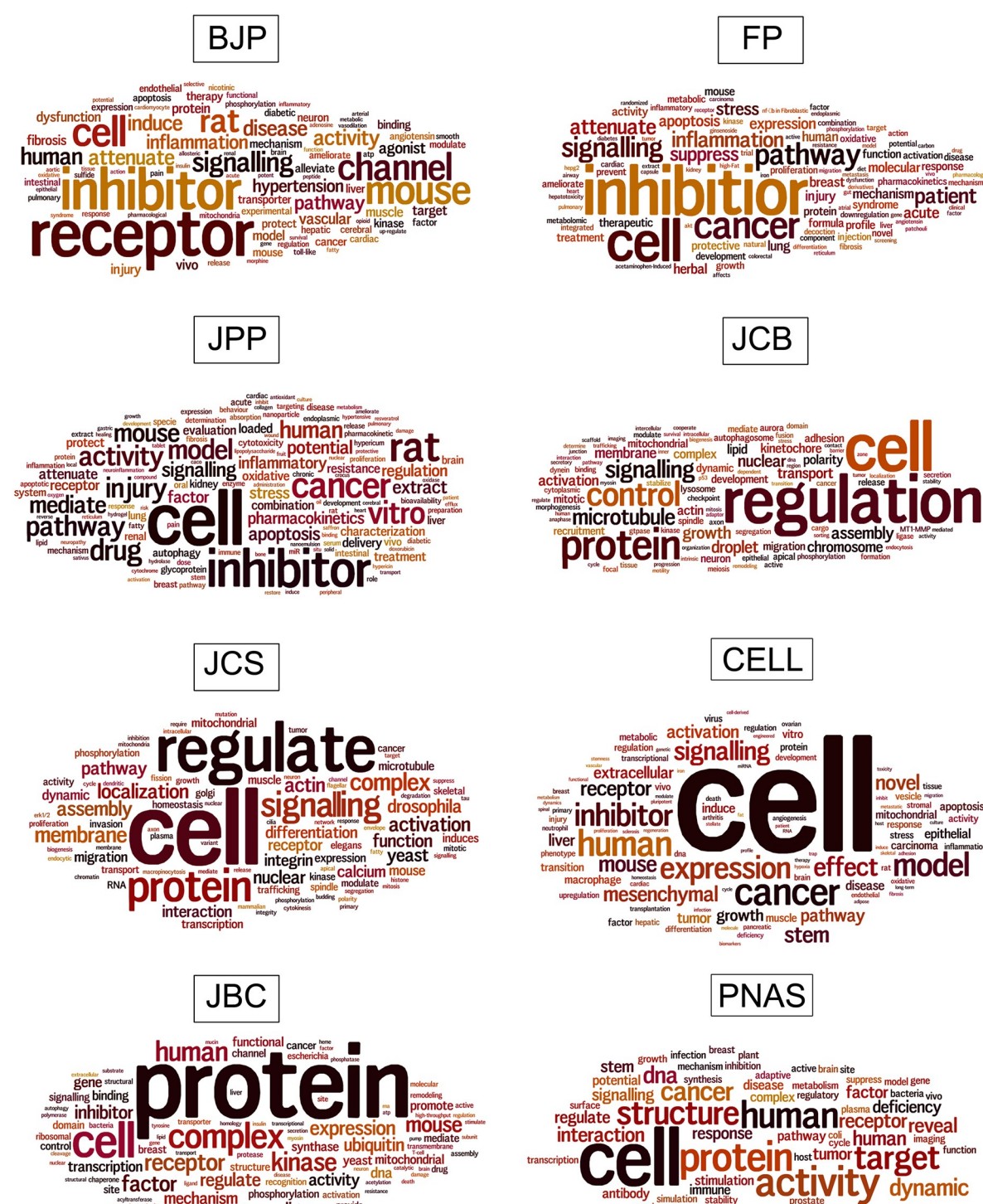

**Fig 5. Comparative analysis of the vocabulary of biomedical sciences journals.** Word clouds were generated using the titles of articles. The clouds give greater prominence to words that appear more frequently in the source text. Data were collected from articles published in 2019 in the biomedical sciences journals: British Journal of Pharmacology (BJP), Journal of Pharmacy and Pharmacology (JPP), Frontiers in Pharmacology (FP), Journal of Cell Biology (JCB), Journal of Cell Science (JCS), Cells (CELL), Journal of Biological Chemistry (JBC) and Proceedings of the National Academy of Sciences (PNAS). N = 200 articles analyzed in each scientific journal.

conventional fluorescence (Flu), confocal fluorescence (Conf), super resolution (SRes), and live cell imaging (Live); and 3 types of electron microscopy techniques: transmission (TEM), scanning (SEM) and cryo-EM (CrEM).
(DOCX)

**S2 File. Frequency of words used in the title of articles from eight biomedical sciences journals.** British Journal of Pharmacology (BJP), Journal of Pharmacy and Pharmacology (JPP), Frontiers in Pharmacology (FP), Journal of Cell Biology (JCB), Journal of Cell Science (JCS), Cells (CEL), Journal of Biological Chemistry (JBC) and Proceedings of the National Academy of Sciences (PNAS). The numbers at the left column of each journal refers to the number of times that each word appeared in the title of the articles (N = 200) from this journal in 2019.
(DOCX)

## Acknowledgments

We thank Professor Luis Eduardo M. Quintas for his careful revision of the manuscript.

## Author Contributions

**Conceptualization:** Claudia Mermelstein.

**Formal analysis:** Agatha M. Reigoto, Sarah A. Andrade, Marianna C. R. R. Seixas, Manoel L. Costa, Claudia Mermelstein.

**Funding acquisition:** Claudia Mermelstein.

**Methodology:** Claudia Mermelstein.

**Project administration:** Claudia Mermelstein.

**Supervision:** Claudia Mermelstein.

**Writing – original draft:** Claudia Mermelstein.

**Writing – review & editing:** Claudia Mermelstein.

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
