## [Decision Letter · Decision Letter 0]

30 Nov 2020

PONE-D-20-26569

How is microscopy used in pharmacology research?

PLOS ONE

Dear Dr. Mermelstein,

Thank you for submitting your manuscript to PLOS ONE. After careful consideration, we feel that it has merit but does not fully meet PLOS ONE’s publication criteria as it currently stands. Therefore, we invite you to submit a revised version of the manuscript that addresses the points raised during the review process.

We look forward to receiving your revised manuscript.

Kind regards,

Yuval Garini, Ph.D.

Academic Editor

PLOS ONE

Journal Requirements:

Additional Editor Comments (if provided):

The manuscript presents a study on the use of microscopy in the fields of pharmacology and cell biology as a control. Three journals are allocated and studied, two in pharmacology and one in cell biology.

Different microscopy methods were analyzed: bright field, phase contrast, DIC, polarization, conventional fluorescence, confocal, TEM and SEM.

The results show different volume of usage in these two fields. In addition, common words were tested in the titles and found ‘Cell’ to be dominant in both fields.

The work is interesting, and the subject is important, but there are major issues that should be corrected and added before it can be published.

Major comments:

Except for the reviewer comments, please refer to the following comments as well.

1. Some important microscopy methods are not included in the subjects that are tested, including Super resolution microscopy that lately immerged as a very important method.

2. Although the issue of ‘in vivo’ and ‘live cell’ are mentioned in the introduction, these are important issues that should also be part of the study. Otherwise, we are left with partial knowledge, as it is not clear what the usage of the microscopy is.

3. It seems that two journals are not enough for the study of a discipline, and definitely a single journal for the control cell biology discipline (with much less articles relative to the pharma discipline number or articles).

4. For the pharmaceutical analysis, it is important to know what kind of study is described in the manuscript. One can assume that studying histological sections will necessitate microscopy, while studying drug tablets not necessarily requires microscopy. Mixing these together may bias the results and certainly the conclusions.

5. In relation to the previous comment, it seems that the value of general statement “use of microscopy in pharmaceutical” is not very high, as it depends what is the actual study. The authors should refine their analysis accordingly.

6. It is important to correlate the use of microscopy with the research theme – such as tablets – tissues – drug efficiency – biomarkers and so on. A ‘comparative analysis” (line 172) similar to that, but for subjects of matter, seemed to be very important.

7. One of the conclusions is: “we can conclude that the pharmacology research field could gain novel horizons by including 215 new microscopy techniques in their studies.”

It does not seem to be a relevant conclusion, definitely not based on the study that is described. For such a statement to be true, one has to explore the type of study, the methods that are used, and compare similar studies that are done with or without microscopy.

That is why point 6 is so important – The use of the technique depends on the application and the subject of study. Microscopy is definitely a great method, but its use, just like any other method, depends on the need.

8. As another comment related to points 6-7, it will be interesting to have a 'negative control' that uses a subject like biochemistry ot other, in order to teat the use of microscopy. THe point to emphasize is that the use of a method is strongly correlated with the need.

Reviewers' comments:

Reviewer's Responses to Questions

**Comments to the Author**

1. Is the manuscript technically sound, and do the data support the conclusions?

Reviewer #1: No

2. Has the statistical analysis been performed appropriately and rigorously? 

Reviewer #1: Yes

3. Have the authors made all data underlying the findings in their manuscript fully available?

Reviewer #1: Yes

4. Is the manuscript presented in an intelligible fashion and written in standard English?

Reviewer #1: Yes

5. Review Comments to the Author

Reviewer #1: This work studies the use of microscopy in pharmacological research. To do this, the authors propose to do a bibliometrics analysis to review scientific publications during the year 2019 on two pharmacology journals (BJP and FP) and one cell biology journal for control.

The following data is recollected

- Whereas microscopy techniques are used.

- Which technique was used, single or combined with others.

- The difference in the subject based on relative frequency of words in the title.

The authors analyze the data a conclude that microscopy techniques are infra utilized in pharmacology.

While I believe the topic is interesting, the analyses performed fail to support the claims make by the authors. A big rewrite must be done so the authors clarify their claims and analyses. As it is, It is not written clearly enough to be accesible and new results should be included to streghen the claims. Therefore I can not recommend it for publication.

Some mayor issues:

- In 211 it is stated that cell biology and pharmacology fields have completely different uses of microscopy techniques”. I do not find clear if the techniques are used for different purposed or it refers that microscopy techniques are more commonly used in cell biology.

- The usage of microscopy is compared using word cloud in the titles in section 192. The use of “cell”, “protein” and “receptor” but no further analysis is given. It do not feel that it is sufficient criteria to state “that differences in microscopy usage cannot be attributed to differences in the subject of study “. Also there is diferent spellings for some words (like signaling/signalling) that can be observed in the word clouds that it is not properly explained.

- No specific reasons are given for using such a small sample of journals (3), even if the number of scientific articles is big enough to guarantee that the analysis was not performed by hand.

-It would be clarifying to add the results to which techniques are mostly use together and if there is any reason for these combinations.

- Finally, the number of references is small and fails to introduce the previous work done in this field. The authors mention the combination of different techniques but fail to cite examples where pharmacology research has improved by the use of microscopy. For example:

Usaj MM, Styles EB, Verster AJ, Friesen H, Boone C, Andrews BJ. High-Content Screening for Quantitative Cell Biology. Trends Cell Biol. 2016;26(8):598–611.

Finally some typos and minor issues

- The results section presents the data in an unhelpful manner. Some restructuring must be done in this section.

-Line 48 How important it is to … -> How important is it to look

-Line 102 Provide Wordle Software as a citation.

6. PLOS authors have the option to publish the peer review history of their article (what does this mean?). If published, this will include your full peer review and any attached files.

Reviewer #1: No

---

## [Author Response · Author response to Decision Letter 0]

31 Dec 2020

Dear Dr. Yuval Garini,

I submitted the manuscript entitled "How is microscopy used in pharmacology research?" (PONE-D-20-26569) for publication in PLOS ONE. The reviewers of my manuscript recommended major revision. I made the modifications in the manuscript and in its figures and I included a point-by-point response to the reviewer’s comments, which were useful to improve the data presentation and interpretation. The whole manuscript text was revised and rewritten, and new figures were added. We decided to change the title of the manuscript to “A comparative study on the use of microscopy in pharmacology and cell biology research” because of the addition of new data. I thank the Referees for their careful and appropriate analysis. The detailed corrections are as follows.

Editor comments:

The manuscript presents a study on the use of microscopy in the fields of pharmacology and cell biology as a control. Three journals are allocated and studied, two in pharmacology and one in cell biology. Different microscopy methods were analyzed: bright field, phase contrast, DIC, polarization, conventional fluorescence, confocal, TEM and SEM. The results show different volume of usage in these two fields. In addition, common words were tested in the titles and found ‘Cell’ to be dominant in both fields. The work is interesting, and the subject is important, but there are major issues that should be corrected and added before it can be published.

FEITO 1 - Some important microscopy methods are not included in the subjects that are tested, including Super resolution microscopy that lately immerged as a very important method. Although the issue of ‘in vivo’ and ‘live cell’ are mentioned in the introduction, these are important issues that should also be part of the study. Otherwise, we are left with partial knowledge, as it is not clear what the usage of the microscopy is.

Author’s response – We now included in the new version of the manuscript “super resolution microscopy” and “live cell/in vivo microscopy”, along with nine other microscopy methods (bright field, phase contrast, differential interference contrast, polarization, conventional fluorescence, confocal fluorescence, transmission and scanning electron microscopy, and cryoEM).

FEITO 2 - It seems that two journals are not enough for the study of a discipline, and definitely a single journal for the control cell biology discipline (with much less articles relative to the pharma discipline number or articles).

Author’s response – We now included data from 8 leading scientific journal from the pharmacology, cell biology, biochemistry, and general biomedical sciences fields. The selected journals were British Journal of Pharmacology (BJP), Journal of Pharmacy and Pharmacology (JPP), Frontiers in Pharmacology (FP), Journal of Cell Biology (JCB), Journal of Cell Science (JCS), Cells (CEL), Journal of Biological Chemistry (JBC) and Proceedings of the National Academy of Sciences (PNAS).

FEITO 3 - For the pharmaceutical analysis, it is important to know what kind of study is described in the manuscript. One can assume that studying histological sections will necessitate microscopy, while studying drug tablets not necessarily requires microscopy. Mixing these together may bias the results and certainly the conclusions. In relation to the previous comment, it seems that the value of general statement “use of microscopy in pharmaceutical” is not very high, as it depends what is the actual study. The authors should refine their analysis accordingly. It is important to correlate the use of microscopy with the research theme – such as tablets – tissues – drug efficiency – biomarkers and so on. A ‘comparative analysis” (line 172) similar to that, but for subjects of matter, seemed to be very important.

Author’s response – We now attempted to correlate the use of microscopy with the research theme of each article. Since the themes can be diverse, we focus on analyzing the use of cell cultures, since it is reasonable to expect that microscopes are needed for the proper visualization of cells because of their dimensions. In articles that used cell cultures, we analyzed the percentage of microscopy figures in relation to the total number of figures of the articles. We thank the reviewer for this important suggestion that highly improved our manuscript.

FEITO 4 - One of the conclusions is: “we can conclude that the pharmacology research field could gain novel horizons by including new microscopy techniques in their studies.” It does not seem to be a relevant conclusion, definitely not based on the study that is described. For such a statement to be true, one has to explore the type of study, the methods that are used, and compare similar studies that are done with or without microscopy. That is why point 3 is so important – The use of the technique depends on the application and the subject of study. Microscopy is definitely a great method, but its use, just like any other method, depends on the need.

Author’s response – We now revised the whole manuscript, including the Conclusions, to incorporate new data on the use of cell cultures together with microscopy. In this way we tried to separate the articles in two types: research at the cellular level or at the whole animal/human level.

FEITO 5 - As another comment related to points 3-4, it will be interesting to have a 'negative control' that uses a subject like biochemistry or other, in order to treat the use of microscopy. The point to emphasize is that the use of a method is strongly correlated with the need.

Author’s response – As I explained above, we now included data from 8 leading scientific journal from the pharmacology, cell biology, biochemistry (as a negative control), and general biomedical sciences fields. I think that these new results have greatly improved our manuscript.

Reviewer #1:

FEITO 1 - This work studies the use of microscopy in pharmacological research. To do this, the authors propose to do a bibliometrics analysis to review scientific publications during the year 2019 on two pharmacology journals (BJP and FP) and one cell biology journal for control. The following data is recollected

- Whereas microscopy techniques are used.

- Which technique was used, single or combined with others.

- The difference in the subject based on relative frequency of words in the title.

The authors analyze the data a conclude that microscopy techniques are infra utilized in pharmacology. While I believe the topic is interesting, the analyses performed fail to support the claims made by the authors. A big rewrite must be done so the authors clarify their claims and analyses. As it is, it is not written clearly enough to be accessible and new results should be included to strengthen the claims. Therefore, I cannot recommend it for publication.

Author’s response – We now included data from 8 leading scientific journal from the pharmacology, cell biology, biochemistry, and general biomedical sciences fields. The selected journals were British Journal of Pharmacology (BJP), Journal of Pharmacy and Pharmacology (JPP), Frontiers in Pharmacology (FP), Journal of Cell Biology (JCB), Journal of Cell Science (JCS), Cells (CEL), Journal of Biological Chemistry (JBC) and Proceedings of the National Academy of Sciences (PNAS); and data from 11 types of optical and electron microscopy techniques: bright field, phase contrast, differential interference contrast (DIC), polarization, conventional fluorescence, confocal fluorescence, super resolution, live cell imaging, transmission and scanning electron microscopy (TEM and SEM, respectively), and cryoEM. The whole manuscript text was revised and rewritten to strengthen our claims and analysis.

FEITO 2 - In 211 it is stated that cell biology and pharmacology fields have completely different uses of microscopy techniques. I do not find clear if the techniques are used for different purposed or it refers that microscopy techniques are more commonly used in cell biology.

Author’s response – We now changed this sentence as well as we reviewed the whole manuscript.

FALTA 3 - The usage of microscopy is compared using word cloud in the titles in section 192. The use of “cell”, “protein” and “receptor” but no further analysis is given. It does not feel that it is sufficient criteria to state “that differences in microscopy usage cannot be attributed to differences in the subject of study”. Also, there is different spellings for some words (like signaling/signalling) that can be observed in the word clouds that it is not properly explained.

Author’s response – We revised all the words present in the word clouds to resolve the different spellings problem (American versus English language). We also reviewed the word cloud section.

FEITO 4 - No specific reasons are given for using such a small sample of journals (3), even if the number of scientific articles is big enough to guarantee that the analysis was not performed by hand.

Author’s response – As I explained above, we now included data from 8 leading scientific journal from the pharmacology, cell biology, biochemistry, and general biomedical sciences fields. The selected journals were British Journal of Pharmacology (BJP), Journal of Pharmacy and Pharmacology (JPP), Frontiers in Pharmacology (FP), Journal of Cell Biology (JCB), Journal of Cell Science (JCS), Cells (CEL), Journal of Biological Chemistry (JBC) and Proceedings of the National Academy of Sciences (PNAS); and data from 11 types of optical and electron microscopy techniques: bright field, phase contrast, differential interference contrast (DIC), polarization, conventional fluorescence, confocal fluorescence, super resolution, live cell imaging, transmission and scanning electron microscopy (TEM and SEM, respectively), and cryoEM. The whole manuscript text was revised and rewritten to strengthen our claims and analyses.

FALTA 5 -It would be clarifying to add the results to which techniques are mostly use together and if there is any reason for these combinations.

Author’s response – We now included a new supplementary figure (Table 1) with the information related to which techniques are mostly used together and we discuss some of these combinations.

FEITO 6 - Finally, the number of references is small and fails to introduce the previous work done in this field. The authors mention the combination of different techniques but fail to cite examples where pharmacology research has improved by the use of microscopy. For example: Usaj MM, Styles EB, Verster AJ, Friesen H, Boone C, Andrews BJ. High-Content Screening for Quantitative Cell Biology. Trends Cell Biol. 2016;26(8):598–611.

Author’s response – We now included more references to introduce the previous work done in this field.

7 - Finally, some typos and minor issues

FALTA - The results section presents the data in an unhelpful manner. Some restructuring must be done in this section.

Author’s response – The Results section of the new version of the manuscript was restructured to make our results easily understandable.

FEITO -Line 48 How important it is to … -> How important is it to look

Author’s response –

FEITO -Line 102 Provide Wordle Software as a citation.

Author’s response – We now provided a citation for Wordle software.

We hope that the modifications made in the new version of the manuscript have properly addressed the criticism and suggestions made by the referee, and that the improvements made in the manuscript will be sufficient for its publication in PLOS ONE.

I would like to state that all listed authors qualify for authorship and agreed in the submission of the manuscript. The final version of the manuscript has been seen and approved by all coauthors. The authors declare that they have no conflict of interest.

With kind regards,

Claudia Mermelstein

---

## [Editor Report · Decision Letter 1]

4 Jan 2021

PONE-D-20-26569R1

A comparative study on the use of microscopy in pharmacology and cell biology research

PLOS ONE

Dear Dr. Mermelstein,

Thank you for submitting your revised manuscript to PLOS ONE.

After careful consideration, we feel that it has merit but does not fully meet PLOS ONE’s publication criteria as it currently stands, pending minor revisions. Therefore, we invite you to submit a revised version of the manuscript that addresses the points raised during the review process.

All the comments of the reviewers were carefully answered as requested, but there are still minor revisions and suggestions as mentioned below.

We look forward to receiving your revised manuscript.

Kind regards,

Yuval Garini, Ph.D.

Academic Editor

PLOS ONE

Editor Comments:

The manuscript is now improved according to all the comments and requests by the reviewers.

There are still few issues to be refined as listed below.

1. PNAS was one of the journals that was evaluated. Note however, that PNAS is a general journal that publish articles in many subjects that are outside the scope of cell biology, or even biology. Consider how to take this into account (i.e. when quoting in page 8 the percentage of OM, the value is lower, 32%, but I believe that if this will be normalized by the percentage of bio-related publications from all publications, the number will become more similar to that of the cell-bio journals).

2. Page 8: “Optical microscopy (OM) was much more used than electron microscopy (EM), probably because EM is more labor-intensive than OM”:

There are many reasons for the broader use of optical microscopy rather than EM, except for the labor issue. It is also because it is faster, possible to label multiple probes simultaneously, works on live cells, the structure is not damaged during the preparation, it is fully quantitative, and even though the resolution is not as good, it is still good enough to provide sub-cellular information. Most likely, it is more accurate to claim that EM is used only where very high resolution is needed, and OM cannot provide that resolution.

3. Page 9: “and an image of the object of study should be almost a requirement for these studies”.

It is not clear where is this fact coming from? Please check the argument, and if necessary, please provide a reference.

4. Page 10 lines 222: Cryo-EM is most likely being used for basic-science studies, which explains the low-percentage use.

5. I may have missed it, but it will be useful to have a table of words usage, the source for the word-cloud images. I t can be a supplement table.

6. Also, it will be very helpful to organize all the relevant factual numbers in one table: number of journals, number of articles in each one. Also add N=… to the figures or figure captions

---

## [Author Response · Author response to Decision Letter 1]

5 Jan 2021

Dear Dr. Yuval Garini,

I submitted the manuscript entitled "A comparative study on the use of microscopy in pharmacology and cell biology research" (PONE-D-20-26569R1) for publication in PLOS ONE. After careful consideration, it was considered that the manuscript has merit but does not fully meet PLOS ONE’s publication criteria as it currently stands, pending minor revisions. Therefore, I made the modifications in the manuscript and in its figures and I included a point-by-point response to the reviewer’s comments, which were very useful to improve the data presentation and interpretation. New figures were added. I thank the Editor for his careful and appropriate analysis.

Editor comments:

The manuscript is now improved according to all the comments and requests by the reviewers. There are still few issues to be refined as listed below.

1 - PNAS was one of the journals that was evaluated. Note however, that PNAS is a general journal that publish articles in many subjects that are outside the scope of cell biology, or even biology. Consider how to take this into account (i.e. when quoting in page 8 the percentage of OM, the value is lower, 32%, but I believe that if this will be normalized by the percentage of bio-related publications from all publications, the number will become more similar to that of the cell-bio journals).

Author’s response – We thank the editor for this important comment. We now analyzed the percentage of OM use in bio-related (biological sciences section of the journal) articles from PNAS. In fact, the percentage of OM in all PNAS’s articles was 32%, whereas the percentage of OM only in bio-related articles is 47%. So, we incorporated this new analysis in the new version of the manuscript.

2 - Page 8: “Optical microscopy (OM) was much more used than electron microscopy (EM), probably because EM is more labor-intensive than OM”. There are many reasons for the broader use of optical microscopy rather than EM, except for the labor issue. It is also because it is faster, possible to label multiple probes simultaneously, works on live cells, the structure is not damaged during the preparation, it is fully quantitative, and even though the resolution is not as good, it is still good enough to provide sub-cellular information. Most likely, it is more accurate to claim that EM is used only where very high resolution is needed, and OM cannot provide that resolution.

Author’s response – We agree with the editor that there are many reasons for the broader use of optical microscopy rather than EM, except for the labor issue. So, we changed this sentence accordingly.

3 - Page 9: “and an image of the object of study should be almost a requirement for these studies”. It is not clear where is this fact coming from? Please check the argument, and if necessary, please provide a reference.

Author’s response – We rephrased this sentence since it was not clear.

4 - Page 10 lines 222: Cryo-EM is most likely being used for basic-science studies, which explains the low-percentage use.

Author’s response – We added this sentence to our manuscript, and we thank the editor for this suggestion.

5 - I may have missed it, but it will be useful to have a table of words usage, the source for the word-cloud images. I t can be a supplement table.

Author’s response – We now included a new Table (supplementary Table 2) in the manuscript with all the words and their frequencies used for the generation of the word clouds.

6 - Also, it will be very helpful to organize all the relevant factual numbers in one table: number of journals, number of articles in each one. Also add N=… to the figures or figure captions.

Author’s response – We now included a new Table in the manuscript containing all the study’s relevant factual numbers (number of journals, journal’s abbreviations, number of articles in each one) and we also added to all the figure captions the number of articles (N) used for each journal.

We hope that the modifications made in the new version of the manuscript have properly addressed the criticism and suggestions made by the referee, and that the improvements made in the manuscript will be sufficient for its publication in PLOS ONE.

I would like to state that all listed authors qualify for authorship and agreed in the submission of the manuscript. The final version of the manuscript has been seen and approved by all coauthors. The authors declare that they have no conflict of interest.

With kind regards,

Claudia Mermelstein

---

## [Editor Report · Decision Letter 2]

8 Jan 2021

A comparative study on the use of microscopy in pharmacology and cell biology research

PONE-D-20-26569R2

Dear Dr. Mermelstein,

We’re pleased to inform you that your manuscript has been judged scientifically suitable for publication and will be formally accepted for publication once it meets all outstanding technical requirements.

Kind regards,

Yuval Garini, Ph.D.

Academic Editor

PLOS ONE

Additional Editor Comments (optional):

The authors have answered all the comments, and the manuscript is now ready for publication.

Well done!

---

## [Editor Report · Acceptance letter]

12 Jan 2021

PONE-D-20-26569R2 

A comparative study on the use of microscopy in pharmacology and cell biology research 

Dear Dr. Mermelstein:

I'm pleased to inform you that your manuscript has been deemed suitable for publication in PLOS ONE. Congratulations! Your manuscript is now with our production department. 

Kind regards, 

on behalf of

Prof. Yuval Garini 

Academic Editor

PLOS ONE